# Grade and Tonnage Comparison of Anthropogenic Raw Materials and Ores for Cu, Zn, and Pb Recovery

Eirik Nøst Nedkvitne *, Dag Øistein Eriksen and Jon Petter Omtvedt

Department of Chemistry, University of Oslo, 0371 Oslo, Norway
* Correspondence: e.n.nedkvitne@kjemi.uio.no

**Abstract:** Primary metal production operates with large tonnages and takes advantage of economies of scale. Metal recycled from low-value waste streams, competing in the same global metal market as primary production, will be more competitive by also taking advantage of up-scaling. However, an overview of metal tonnages in low-value waste streams to see upscaling potential needs to be provided in the literature. In response, this study provides estimates of copper, zinc, and lead metal tonnages in waste incineration ash—A major waste stream going to landfills. Metal concentrations and tonnages are compared to tonnages and concentration grades found in ores. Copper, zinc, and lead concentration averages are about 3–5 times lower in ash compared to the worldwide average head grade of ores. Tonnages of metal in the ash generated from waste incineration in European countries bordering the Baltic and the North Sea are about 1/3 of mining metal output from Sweden, a leading mining country in the region. Therefore, incineration ash should be considered a significant potential Cu, Zn, and Pb metal source.

**Keywords:** MSW; MSWI; fly ash; bottom ash; copper; zinc; lead; low value waste stream; Europe; recycling; waste management





## 1. Introduction

Small metal tonnages often constrain economically feasible recycling from low-value waste streams. Such recycling can be costly compared to primary metal productions, which have the advantages of economies of scale, easier transport logistics, and more homogenous raw material streams. Today, recycling is often limited to high-value metal waste streams, such as waste from electrical and electronic equipment and vehicle scrap. It is economically feasible due to the relatively easy sorting, remelting, and refining procedures. In contrast, metal recycling from low-value waste streams faces more technical challenges, similar to those facing primary metal production. Moreover, the economic benefits of large annual primary metal production tonnages often outcompete recycling from small annual tonnages of low-value waste.

Substantial amounts of anthropogenic copper, zinc, and lead are lost in waste flows going to incineration. Metals are sorted out from the waste, both before and after incineration. However, conventional physical separation methods cannot recover metals embedded in products or small-size particles [1]. Fine particle fractions go to landfills [2]. This applies to municipal solid waste incineration (MSWI) fly ash and fine particle fractions of MSWI bottom ash [3]. MSWI fly ash and MSWI bottom ash are in this paper further referred to as fly ash and bottom ash. The general term for the two is referred to as "waste ash". High concentration of heavy metals in fly ash is classified as a hazardous waste. A waste stream must be carefully treated before being deposited in special hazardous waste landfills. Hence, recycling metals from fly ash has received considerable attention for solving waste management challenges with fly ash [4], particularly since large tonnages are generated. Figure 1 shows tonnages of waste incinerated in central and northern European countries.

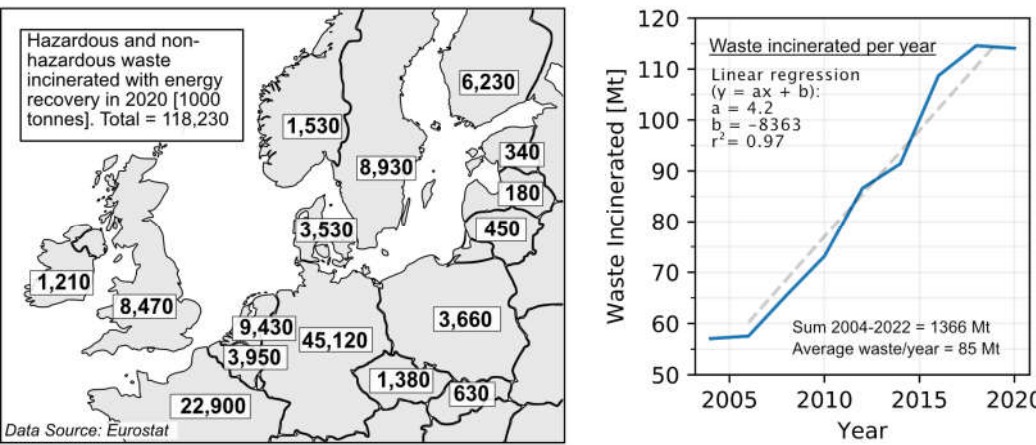

**Figure 1.** (**Left**): Map of northern Europe showing municipal solid waste tonnages (in thousands of tonnes) incinerated by country in 2020. (**Right**): MSW tonnages incinerated with energy recovery by year for the countries shown in the left figure. Eurostat is the source of data [5].

Today, zinc metal is recovered from one incineration plant in Switzerland using acid leaching and solvent extraction technology [6,7]. However, few other plants have implemented the technology due to high costs. One opportunity to reduce costs is processing larger ash tonnages and exploiting economies of scale at a centralized processing plant. Hence, metal recycling may be more competitive in a global metal market and contribute to meeting metal recycling goals [8]. Mapping metal tonnages in ash is essential to assessing the potential for large-scale processing. This study aims to estimate annual tonnages and concentrations of copper, zinc, and lead in waste ash, which can be compared to primary production data for comparison. This contributes to building a foundation for the qualitative assessment of an economically viable metal recycling process from waste ash competing with primary metal production.

## 2. Method

Waste ash copper, zinc, and lead concentrations are compared to head grades of ores rich in the same metals. Data on head grades were gathered from publicly available MDO data [9]. Fly ash concentrations are gathered from the Langøya data set (XRF of 895 ash samples) [10]. The data is available for download in supporting information. A model for estimating metal tonnages in fly ash and fines from bottom ash was built. Fines from bottom ash refer to the fine particle fraction of bottom ash. The model is based on Monte Carlo methods to consider uncertainties in input variables. Concentrations in fly ash for the tonnage estimate are based on the Langøya data set. Metal concentrations in the fines of bottom ash are assumed to be equal to those in fly ash, though literature indicates slightly higher concentrations of copper and lower concentrations of zinc [11–14]. For the metal tonnage estimates, four different cases are considered.

The first case comprises the tonnage of fly ash landfilled at Langøya annually. In 2021, 370,000 tonnes of fly ash originating from Sweden, Denmark, Norway, Ireland, and Lithuania were treated and landfilled there. Case 2 comprises fly ash as in case 1, but in addition, there are estimated amounts of fines in bottom ash originating from the same incinerators as the landfilled fly ash. The estimate is based on the ratio of fly ash to bottom ash generation from waste incineration and fractions of fines in bottom ash. The cut-off particle size for fines in bottom ash is set at 0.5 mm, as state-of-the-art physical separation methods cannot separate out smaller metallic particles than this [1]. The mass fraction of particles under 0.5 mm is derived from the average particle size distribution of bottom ash [1] (see Figure 2). This leads to a fraction making up 20% of the total bottom ash mass. Chemical selective separation methods, on the other hand, may be more appropriate than physical separation methods for metal recovery of particles larger than 0.5 mm. In addition, due to technological hindrances in treating fine particle fraction in Europe, 54 wt% of treated bottom ash went

to landfills in 2018 [2]. This is considerably more than the estimated 20% derived from a 0.5 mm cut-off particle size. Tonnages for bottom ash fines in this study may therefore be an underestimation. Case 3 estimates fly ash tonnages from statistics from Eurostat [5] of waste tonnages incinerated with energy recovery from the northern European countries shown in Figure 1. Case 4 also considers estimated amounts of fines from bottom ash. The equations used for the estimates are as follows:

$$M_m = M_{FA} \, c_m \tag{1}$$

$$M_m = M_{FA} \, c_m + (f_{BA} \, /f_{FA}) \, M_{FA} \, c_m \tag{2}$$

$$M_m = M_{MWS} \, f_{FA} \, c_m \tag{3}$$

$$M_m = M_{MWS} \, f_{FA} \, c_m + M_{MWS} \, f_{BA} \, f_{fines} \, c_m \tag{4}$$

Equations (1)–(4) are used for cases 1, 2, 3, and 4, respectively. $M_m$, $M_{FA}$, $M_{MSW}$, $c_m$, $f_{BA}$, $f_{FA}$, and $f_{fines}$ stand for annual tonnage of metal, annual tonnage of fly ash, annual tonnage of municipal solid waste, concentration of metal in ash, fraction of bottom ash generated by incineration, fraction of fly ash generated by incineration, and percentage of fines in bottom ash, respectively. The input variables' descriptions and values, uncertainties (standard deviation), and references are presented in Table 1.

**Table 1.** Variables for the tonnage estimation model.

| Variable, Symbol, [Unit] | Value | SD | Source |
|---|---|---|---|
| Cu concentration, $c_{Cu}$, [wt%] | 0.27 | 0.04 | Langøya data set |
| Zn concentration, $c_{Zn}$, [wt%] | 1.09 | 0.04 | Langøya data set |
| Pb concentration, $c_{Pb}$, [wt%] | 0.26 | 0.02 | Langøya data set |
| FA generated per MSW, $f_{FA}$, [%] | 3.00 | 0.2 | [15] |
| BA generated per MSW, $f_{BA}$, [%] | 27.5 | 2 | [15] |
| Percentage fines in BA, $f_{fines}$, [%] | 20.00 | 3 | [1] |
| MSWI fly ash at Langya, $M_{FA}$, [t] | 350,000 | – | NOAH AS [10] |
| MSW in northern Europe, $M_{MSW}$, [t] | 118,200,000 | – | Eurostat [5] |

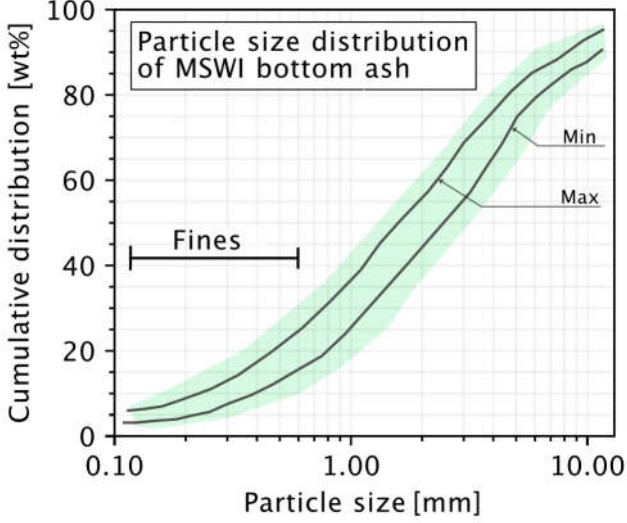

**Figure 2.** Typical minimum and maximum cumulative particle size distribution function of bottom ash [1], which is used to estimate tonnages of fines in bottom ash. The figure is adapted from [17], with permission from Elsevier, 2023.

Annual tonnages are compared with reference annual output tonnages from the mining industry. Sweden has one of the largest mining industries in Europe, and tonnages of mined metal outputs are used as a reference. The country has large mines for copper,

zinc, and lead. Their annual metal mining outputs are 100,000 (SWE-1), 250,000 (SWE-2), and 70,000 tonnes (SWE-3) for copper, zinc, and lead, respectively [16]. In addition, the production capacities of Norwegian refineries for zinc and copper are used as tonnage comparisons. Boliden Odda (NOR-1) has a zinc production capacity of 350,000 tonnes, and Glencore Nikkelverk (NOR-2) has a production capacity of 39,000 tonnes of copper. The tonnage estimates and concentration are also compared to selected mines processing low-grade copper ores [9].

## 3. Results

### 3.1. Concentrations

Figure 3 shows boxplots of fly ash concentrations from the Langøya data set and head grade ores from mines reported in public MDO data The concentration of valuable metals is lower in fly ash compared to ore head grades. The mean copper concentration is 3.4 times lower in ash (0.27 wt%) than the average copper head ore grade (0.92 wt%), while the mean zinc concentration is 3.9 times lower (1.09 wt% compared to 4.2 wt%). The difference is even larger for lead, with a 5.3-fold lower mean concentration (0.26 wt% compared to 1.34 wt%). The median concentrations are lower than average head ore grades and concentrations in ash. This reflects the fact that portions of both raw materials have higher grades.

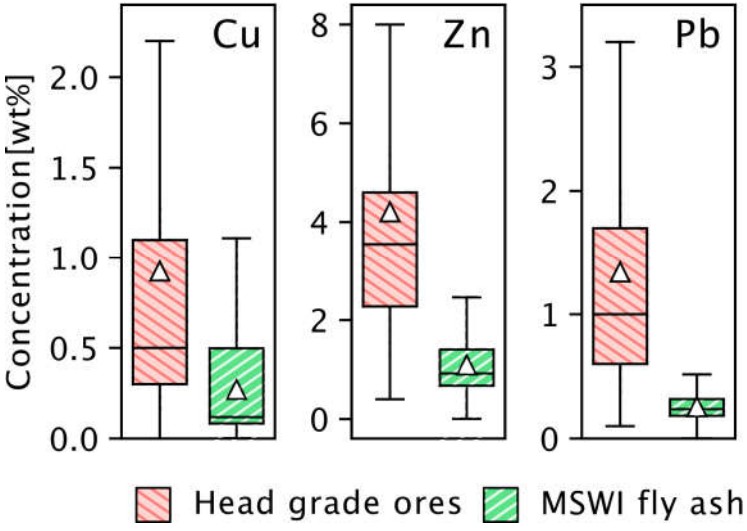

**Figure 3.** Boxplots of head grade ores (red) (number of mines: nCu = 219, nZn = 64, nPb = 45) and fly ash concentrations (green) (895 ash samples from 26 different plants located in Sweden and Norway). Mean concentrations are marked with a triangle. Source: 2017–2022 MDO Data Online Inc. (public data) [9] and Langøya data set [18].

The copper mean concentration of head grade ores from the MDO dataset was higher than the stated average copper ore grade, equaling 0.76%, of today's mining projects utilizing milling/flotation-based technology [19]. In addition, the global average copper ore grade is stated to be as low as 0.62% [20]. Taking this into consideration, the copper concentration in fly ash is only 2.8 and 2.3 times lower compared to the two references mentioned, respectively.

### 3.2. Tonnages

Annual tonnages of ash and metals in ash are presented in Table 2 with the percentages of reference tonnages. According to estimates, annual metal tonnages can be doubled by including fines from bottom ash with fly ash. This comprises case 2 compared to case 1 and case 4 compared to case 3. Uncertainties are lowest in case 1, where fly ash landfilled at Langøya is considered. Here, copper and zinc tonnages represent $1.1 \pm 0.1$ and $2.4 \pm 0.4$ percent of the production capacities of the Boliden Odda zinc

refinery and the copper refinery, respectively. The highest annual ash tonnage (case 4) of $10.1 \pm 1.1$ million tonnes of ash is estimated to have 27,300, 110,100, and 25,800 tonnes of copper, zinc, and lead, respectively. These annual tonnages occupy large parts of the Norwegian refinery capacity (70% for copper and 31% for zinc) and are roughly equal to 1/3 of Swedish metal output from mining.

**Table 2.** Annual tonnage estimates of metal in incineration waste ash from four different cases. Cases 1 and 3 include only fly ash, and cases 2 and 4 include fly ash and fines in bottom ash. Cases 1 and 2 comprise ash landfilled at Langøya, while cases 2 and 3 are estimates from MSW generated from countries bordering the Baltic and North Sea. The percentage of reference tonnages from primary production is also presented.

| Case | Variable | Mass Flow [t/Year] | Percentage of SWE1–3 [%] [1] | Percentage of NOR1–2 [%] [2] |
|---|---|---|---|---|
| Case 1: MSWI fly ash processed at Langøya | Ash | 350,000 | – | – |
| | Cu | $900 \pm 100$ | $0.9 \pm 0.1$ | $2.4 \pm 0.4$ |
| | Zn | $3800 \pm 100$ | $1.5 \pm 0.1$ | $1.1 \pm 0.1$ |
| | Pb | $900 \pm 100$ | $1.3 \pm 0.1$ | – |
| Case 2: Case 1 plus related bottom ash fines | Ash | $995,000 \pm 116,000$ | – | – |
| | Cu | $2700 \pm 500$ | $2.7 \pm 0.5$ | $6.9 \pm 1.3$ |
| | Zn | $10,900 \pm 1300$ | $4.3 \pm 0.5$ | $3.1 \pm 0.4$ |
| | Pb | $2500 \pm 400$ | $3.6 \pm 0.5$ | – |
| Case 3: MSWI fly ash from northern Europe | Ash | $3,546,000 \pm 535,000$ | – | – |
| | Cu | $9600 \pm 2000$ | $9.6 \pm 2.0$ | $24.6 \pm 5.2$ |
| | Zn | $38,700 \pm 6000$ | $15.5 \pm 2.5$ | $11.1 \pm 1.8$ |
| | Pb | $9100 \pm 1500$ | $13.0 \pm 2.2$ | – |
| Case 4: Case 3 plus related bottom ash fines | Ash | $10,072,000 \pm 1,113,000$ | – | – |
| | Cu | $27,300 \pm 5100$ | $27.3 \pm 5.2$ | $70.0 \pm 13.3$ |
| | Zn | $110,100 \pm 12,800$ | $44.0 \pm 5.3$ | $31.4 \pm 3.8$ |
| | Pb | $25,800 \pm 3500$ | $36.8 \pm 5.2$ | – |

[1] SWE1 for Cu: Swedish copper mines output (100,000 t/year); SWE2 for Zn: Swedish zinc mines output (250,000 t/year); and SWE3 for Pb: Swedish lead mines output (70,000 t/year). [2] NOR1 for Cu: Glencore Nikkelverk copper production capacity (39,000 t/year); NOR2 for Zn: Boliden Odda zinc production capacity (350,000 t/year).

In 2000, about 60% of the zinc produced worldwide came from mines producing under 100,000 tonnes of zinc annually [21]. For copper, mines producing less than 50,000 tonnes of copper per year account for 16% of global production [21]. By comparing annual metal tonnage estimates from waste ash generated in northern Europe to worldwide mine sizes, waste ash can be seen as a larger zinc resource than a copper resource. Table 3 presents the head ore grades and annual production of selected operating open-pit mines that are processing low-grade copper ores. The head grades are, on average, similar to waste ash concentrations, with 0.3% compared to 0.27%. The average annual production is 63,600 tonnes of copper, which is about twice as much as the annual tonnages of copper in waste ash in northern Europe, according to the highest estimates provided in this study.

Copper and zinc concentrations vary with different types of fly ash [18]. The unequal distribution of different types of ashes landfilled at Langøya and the distribution of ash types generated in northern Europe may be a source of uncertainty in the tonnage estimate. The Langøya data set has about 20% of fly ashes coming from fluidized incinerators, where copper concentrations are higher and zinc concentrations are lower than the average. Therefore, a slight overestimation of copper (and an underestimation of zinc) may have occurred, as fluidized bed incinerators are not as common as grate incinerators. However, copper concentrations in the fines of bottom ash are, in general, higher than fly ash, and zinc is lower. That may compensate for the estimation errors and even result in overestimation.

**Table 3.** Selection of open-pit mines processing low-grade copper ores. Source: 2017–2022 MDO Data Online Inc. (public data) [9].

| Mine | Head Grade | Annual Production |
|---|---|---|
| Mount Milligan Mine (CA) | 0.23% Cu, 0.39 g/t Au | 25,800 t Cu (2016) |
| Aitik Mine (SE) | 0.28% Cu, 0.2 g/t Au, 2 g/t Ag | 99,300 t Cu (2018) |
| Sierrita Mine (US) | 0.23% Cu, 0.02% Mo | 68,900 t Cu (2018) |
| Mount Polley Mine (CA) | 0.38% Cu, 0.3 g/t Au, 0.9 g/t Ag | 6791 t Cu (2018) |
| Pinto Valley Mine (US) | 0.35% Cu, 0.007% Mo | 58,513 t Cu (2021) |
| Constancia Mine (PE) | 0.31% Cu, 89 g/t Mo, 0.07 g/t Au, 3.04 g/t Ag | 122,178 t Cu (2018) |

The cut-off particle size of 0.5 mm for defining bottom ash fines makes the tonnage estimate conservative. Considering that 54% of bottom ash fines go to landfill [2], and assuming the same copper concentration as fly ash in this fraction, the tonnage contribution from bottom ash fines would be doubled. This would result in a 1.6 times higher metal tonnage than the highest estimate presented.

Waste incineration has been ongoing for decades and large tonnages of ash have been landfilled. Subsequently, large tonnages of copper, zinc, and lead can be found in such landfills. According to Eurostat, 1366 million tonnes of waste were incinerated from 2004 to 2020 [5]. Hence, it can be estimated that on average, 6900, 26,000, and 6600 tonnes of copper, zinc, and lead have been incorporated in landfilled fly ash annually (fines in bottom ash not considered). These tonnages may be exploitable stocks for metal recovery in addition to annual waste ash flows. The accessibility of this potential resource should be further studied.

## 4. Discussion

Tonnages and concentration grades indicate the potential for economically feasible metal recycling. However, many other variables are essential and decisive: how easy the metals are to separate, the energy demand of extraction technology, the potential for generation of valuable by-products, the amount of generated waste that needs treatment, the ease of removing contaminants for downstream refining, transport logistics, the amount of technology development, research needed to exploit the raw materials, etc. Comparing waste ash with ores can provide qualitative considerations for using ash as a raw material for metal production. First, as in all recycling cases, environmental impacts from mining are avoided.

Metal recycling from waste ash does not need energy-consuming comminution or shredding (for example, shredding of vehicle scraps). The energy usage for comminution in mining increases as average ore grades deplete [20]. In copper mining, comminution can take up to 30–50% of the energy used to produce the final metal product [22,23]. An economic evaluation of copper extraction from copper mining tailings in Chile states that the tailing needs a grade above 0.41% to be economically viable [24]. This is another waste stream that does not require comminution. The grade is lower than the average copper concentration in fly ash originating from fluidized bed incinerators, which is 0.52% [18].

As fly ashes are classified as hazardous waste, recycling procedures can "detoxify" the ash and improve its properties for landfilling. Hence, recycling metals in waste ash can also be considered a waste management service and help reduce the fill-up rates of special hazardous waste landfills. Reduced cost by avoiding the deposition of untreated fly ash is essential for the economically feasible implementation of the zinc extraction process, FLUWA FLUREC. Economic analyses show that avoiding standard fly ash deposition makes up 61 percent of total savings [7,25]. This demonstrates the importance of the quality of processed ash for landfilling after a metal extraction process. If processed ash deteriorates regarding the metal leachability properties for landfilling, it significantly hinders achieving an economically viable process.

Transport logistics may be more challenging in waste ash processing than in primary metal production, as ash generation is distributed over larger areas. However, today's waste ash management practices, e.g., ash transported to Langøya, show this is practically feasible even without an additional material recovery process. Ash metal recycling offers opportunities for the production of by-products like salt or raw materials for clinker production [4]. A salt extraction plant is now (2023) opening in Sweden using Ash2Salt technology, demonstrating economically viable salt recovery from fly ash [26]. Thus, a metal extraction process should be able to generate valuable by-products like salt.

Metal extraction and mining are capital-intensive industries. Exploiting the economics of scale is therefore important. Zinc and copper mines have increased in size to exploit economies of scale over the past century. This is in order to compensate for increased production costs related to the depletion of ore grades [21]. The Australian copper mining industry had an average return on scale from 1969–1995 of 0.15 [27]. This demonstrates how favorable upscaling has been in the capital-intensive industry. Likewise, capital investments make up the largest share of costs in the economic analysis of implementing the FLUWA-FLUREC process [7,25]. Therefore, scaling up a metal extraction plant may be important to compensate for the large investments. To exploit large waste ash tonnages, a metal extraction process must be robust to process differences in fly ash types. The annual estimated tonnages presented in this study show that it is possible to scale up metal production from fly ash and bottom ash fines to equal sizes as typical mines operating today [21].

Technology for processing ores has been improving for centuries, and processes have been fine-tuned for low production costs. This contrasts with metal extraction from waste ash, where technology must be improved and infrastructure built. In addition, ash has some characteristics different from most common processed ores that need to be considered differently. The fine particle size and complex crystallographic composition may make selective beneficiation techniques difficult. This, in combination with the low-grade, indicates the use of a hydrometallurgical extraction route rather than a pyrometallurgical one. Waste ash has a fine particle size and a non-uniform particle size distribution. This may negatively influence the percolation properties for percolation leaching methods [28]. Pre-agglomeration may be needed. Percolation leaching is often used in large-scale mines of low-grade ores and may be interesting for large-scale processing. Waste ash also has a higher proton exchange capacity than common ores. This can be costly due to the high acid consumption of an acidic extraction route. Waste ash has different contaminants compared to common ores, e.g., the high content of halogens can be difficult to handle in zinc electrowinning plants processing ores. The special waste ash properties must be considered when developing the best possible metal recovery process.

Tonnage estimates show that large-scale operations for metal recovery from fly ash and bottom ash are possible. This will reduce investment costs per metal produced. Lower waste ash grade compared to a higher grade of mineral ores may be compensated by lack of comminution, the potential avoiding conventional hazardous waste ash treatment process, and the ability to generate secondary products (e.g., salts). More efforts should be invested in making metal recovery from waste ash a reality. Many positive indications for possible economically viable processes can be seen when comparing waste ash with ores. Furthermore, metal extraction from waste ash must be seen as a holistic process for metal recovery and the treatment of hazardous waste ash.

## 5. Conclusions

Considerable copper, zinc, and lead tonnages can be found in waste incineration ashes. Yearly flows have tonnages that are comparable to typical annual mine tonnage outputs. Estimates show that fly ash and bottom ash fines from countries bordering the Baltic and North Sea make up about 1/3 of Swedish mining output for copper, zinc, and lead production. However, metal concentrations in ash are lower than head grades from

mines. The concentrations are about 3–5 times lower than the average head grades of worldwide ores.

**Supplementary Materials:** The following supporting information can be downloaded at: https://www.mdpi.com/article/10.3390/resources12030033/s1, the supporting information is consented to the Figure 3: Metal concentration comparison.

**Author Contributions:** Conceptualization, E.N.N.; formal analysis, E.N.N.; investigation, E.N.N.; writing—original draft preparation, E.N.N.; writing—review and editing, E.N.N., D.Ø.E. and J.P.O.; visualization, E.N.N.; supervision, D.Ø.E. and J.P.O.; project administration, D.Ø.E. and J.P.O. All authors have read and agreed to the published version of the manuscript.

**Funding:** This research was funded by the Research Council of Norway, grant number 294543.

**Data Availability Statement:** The data presented in this study are available in supplementary data.

**Acknowledgments:** Kai Erik Ekstrøm, Morten Breinholt Jensen, and Haakon Rui from NOAH AS and the PRICE project collaboration.

**Conflicts of Interest:** The authors declare no conflict of interest.

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
