# Peer review of "Grade and Tonnage Comparison of Anthropogenic Raw Materials and Ores for Cu, Zn, and Pb Recovery"

_resources, doi:10.3390/resources12030033_

Round 1

Reviewer 1 Report

This is a fairly simple paper that attempts to roughly estimate the potential for recovering selected metals from waste incineration residues. The paper does not really generate any new knowledge.

Page 2, line 50. The references [7] and [8] are probably wrong.

In this paragraph, I also miss a reference to the paper by Fellner et al, which deals with a similar topic in much more depth.

J. Fellner, J. Lederer, A. Purgar, A. Winterstetter, H. Rechberger, F. Winter, D. Laner, Evaluation of resource recovery from waste incineration residues - The case of zinc, Waste Management, Volume 37, 2015, Pages 95-103, ISSN 0956-053X, https://doi.org/10.1016/j.wasman.2014.10.010.

Equation 4 and Table 1: Apparently the same metal concentration is assumed for fly ash and bottom ash. In any case, this is grossly incorrect.

Page 5, line 106: The sentence "However, the mean concentrations found in fly ash are substantial." cannot be deduced from the context and Figure 3. Rather, the opposite is the case.

Reviewer 2 Report

Grade and Volume Comparison of Anthropogenic Raw Materials and Ores for Cu, Zn and Pb Recovery

resources-2134899

Review

The authors describe a comparative study on natural and waste resources with focus on the three elements Cu, Zn and Pb. I accepted the review because I am working in the area of (metal) resources from waste. From the content of the submission I am a little bit disappointed because actually nothing new is presented. I do not recommend rejection because I think it might be of interest for some of the journal’s readers after major revision. The authors should follow the specific comments below.

Specific comments

Line 50: There is something wrong with the reference list. After ref 3 in line 44 next reference is 7 and 8 in line 50. And 7 and 8 have nothing to do with zinc recovery in Switzerland. Reference 4 would fit but it is a Ph.D. thesis with 27 MB. As far as I know there are also journal articles on this subject by Gisela Weibel. Ref 4, 5 and 6 are not used in the text.

Line 60: The Langoya data set cannot be found anywhere. The respective information should be presented in a table of new Supporting Information.

Line 60: the term “volume” is misused throughout the whole document. What you mean is mass flow in tons per year. Change accordingly throughout the text.

Line 64: “Metal concentrations in fines of bottom ash are assumed equal to fly ash, though literature indicates higher concentrations of copper and lower concentrations of zinc”. I do not think hat this statement is correct.

Line 79: You mean Eurostat not Eurostate. The tables from Eurostat have specific numbers and labels so that a correct citation is possible. Change accordingly.

Table 1. Use comma as 1000-separator not space.

Line 90-98. This paragraph is confusing in connection with table 2. Replace by

Mass flows are compared with reference volumes from the mining industry. Sweden has one of the largest mining industries in Europe, and volumes of mined metal outputs are used as a reference. The country has large mines for copper, zinc and lead. Their annual metal mining outputs are 100,000 (SWE-1), 250,000 (SWE-2) and 70,000 tonnes (SWE-3) for copper, zinc and lead, respectively [11]. Also, the production capacities of Norwegian refineries for zinc and copper are used as volume comparisons. Boliden Odda (NOR-1) has a zinc production capacity of 350,000 tonnes, and Glencore Nikkelverk (NOR-2)has a production capacity of 39,000 tonnes of copper.

Add refences for the mining data.

Line 107: What is substantial?

Figure 3. Use colors which are better for b/w printing.

Line 119. Replace by Mass flows (see above).

Table 2: Replace Volume by Mass flow. Replace Percentage of ref.1(resp. 2) by ore grade of SWE1-3 or NOR1-2.

Line 149: Add reference to “Eurostat”, see above.

Line 158: Replace usage by demand.

Line 160: contaminants

Line 163: Change to First, as in all recycling cases, environmental impacts from mining is avoided.

Ref. 1 and 3: List all authors even when the list is long.

Ref. 10. Abbreviate first name of author. The doi is for the complete Encyclopedia. Replace by 10.1002/14356007.w28_w02. Follow the citation hints from the publisher: Quicker, P. (2020). Waste, 7. Thermal Treatment. In Ullmann's Encyclopedia of Industrial Chemistry. https://doi.org/10.1002/14356007.w28_w02.

Ref. 11: Add publisher (USGS National Minerals Information Center).

Improve chapter 4 Discussion. There is literature on the Zn recovery in Switzerland (e.g. Fellner, J., Lederer, J., Purgar, A., Winterstetter, A., Rechberger, H., Winter, F., and Laner, D., Evaluation of resource recovery from waste incineration residues – The case of zinc, Waste Manage 37 (2015) 95-103. doi: http://dx.doi.org/10.1016/j.wasman.2014.10.010 and Purgar, A., Winter, F., Blasenbauer, D., Hartmann, S., Fellner, J., Lederer, J., and Rechberger, H., Main drivers for integrating zinc recovery from fly ashes into the Viennese waste incineration cluster, Fuel Process Technol 141, Part 2 (2016) 243-248. doi: http://dx.doi.org/10.1016/j.fuproc.2015.10.003), also including the aspect of cost.

There is so far no discussion on the value of resources. Natural resources might have higher value than resources from waste in terms of exergy, e.g. sulfidic ores versus oxidic waste materials. For a discussion on this aspect see literature by CIRCE in Spain (Alicia Valero, e.g. Calvo, G., Valero, A., and Valero, A., Material flow analysis for Europe: An exergoecological approach, Ecological Indicators 60 (2016) 603-610. doi: http://dx.doi.org/10.1016/j.ecolind.2015.08.005 or Valero, A., Valero, A., and Dominguez, A., Exergy Replacement Cost of Mineral Resources, Journal of Environmental Accounting and Management 1(2) (2013) 147-158.

For copper this aspect was discussed for ores and bottom ash from MSWI (Simon, F.G., and Holm, O., Exergetische Bewertung von Rohstoffen am Beispiel von Kupfer, Chemie Ingenieur Technik 89(1-2) (2017) 108-116. doi: 10.1002/cite.201600089). The impact of improved recycling of metals form bottom ash from MSWI was discussed by Bruno et al. (Bruno, M., Abis, M., Kuchta, K., Simon, F.-G., Grönholm, R., Hoppe, M., and Fiore, S., Material flow, economic and environmental assessment of municipal solid waste incineration bottom ash recycling potential in Europe, J Clean Prod (2021) 128511. doi: 10.1016/j.jclepro.2021.128511).

The authors should discuss their data with these findings from literature and answer the question Where is a realistic “cut-off value” for natural resources and resources from waste?

Reviewer 3 Report

Information on Cu, Zn, and Pb amounts in waster material is useful, and comparison between MSWI and ore is important to consider feasibility of metal recovery from waste material. However, this should be discussed from multiple perspective, and such discussion is not provided in this paper. Therefore, major revision is needed.

1.      It is desirable to show benefit to be derived from the metal recovery and compare the benefits between MSWI fly ash and head grade ores.

2.      In addition to benefit, costs of metal recovery are important. The cost comparison between MSWI fly ash and the ores should be discussed.

3.      There are a variety of metal recovery techniques, and their applicability may depend on chemical composition of MSWI fly ash. Impurities in feedstocks inhibit metal recovery, and the compositional difference between the fly ash and the ore should be discussed to determine efficient metal recovery process.

4.      The description “recycling procedures can “detoxify” ash” (page 6 line 166). This depends on recycling processes. When extraction process using acid or alkaline solution was used for metal recovery, the leachability of hazardous metals in the residual solid possibly increase. It is necessary to discuss the detoxification.

Round 2

Reviewer 2 Report

The manuscript was improved considerably. It can be published after minor revisions.

1.) You added a csv-file with the Langoya data set. I assume this will be available as Supporting information or Supplement. Add to ref (10) and at first appearance in the text that the data are available for download.

2.) Check sentence in line 256/57: “this” instead of “there”?

3.) Ref (6): Add Ph.D. thesis, University of Berne (CH).

4.) Ref. (15) is still not correctly formatted:

Quicker, P. Waste, 7. Thermal Treatment. In ULLMANN'S Encyclopedia of Industrial Chemistry; Wiley-VCH: Weinheim, 2020, https://doi.org/10.1002/14356007.w28_w02.

Author Response

Thank you very much for reviewing the manuscript. We also feel the manuscript was considerably improved after good feedback from reviewers.

We have changed all suggested changes given point by point.

Kindly regards,
Authors of the manuscript

Reviewer 3 Report

Corrections have been made based on the comments. We also understood the responses. The revised manuscript can be accepted.

Author Response

Thank you very much for reviewing the manuscript. We also feel the manuscript was considerably improved after good feedback from you and other reviewers.

Kindly regards,
Authors of the manuscript